# Precision Telemedicine through Crowdsourced Machine Learning: Testing Variability of Crowd Workers for Video-Based Autism Feature Recognition

**DOI:** 10.3390/jpm10030086

**Published:** 2020-08-13

**Authors:** Peter Washington, Emilie Leblanc, Kaitlyn Dunlap, Yordan Penev, Aaron Kline, Kelley Paskov, Min Woo Sun, Brianna Chrisman, Nathaniel Stockham, Maya Varma, Catalin Voss, Nick Haber, Dennis P. Wall

**Affiliations:** 1Department of Bioengineering, Stanford University, 443 Via Ortega, Stanford, CA 94305, USA; peterwashington@stanford.edu (P.W.); briannac@stanford.edu (B.C.); 2Department of Pediatrics (Systems Medicine), Stanford University, 1265 Welch Rd., Stanford, CA 94305, USA; emilie.leblanc@stanford.edu (E.L.); kaiti.dunlap@stanford.edu (K.D.); ypenev@stanford.edu (Y.P.); akline@stanford.edu (A.K.); 3Department of Biomedical Data Science, Stanford University, 1265 Welch Rd., Stanford, CA 94305, USA; kpaskov@stanford.edu (K.P.); minwoos@stanford.edu (M.W.S.); 4Department of Neuroscience, Stanford University, 213 Quarry Rd., Stanford, CA 94305, USA; stockham@stanford.edu; 5Department of Computer Science, Stanford University, 353 Jane Stanford Way, Stanford, CA 94305, USA; mvarma2@stanford.edu (M.V.); catalin@cs.stanford.edu (C.V.); 6School of Education, Stanford University, 485 Lasuen Mall, Stanford, CA 94305, USA; nhaber@stanford.edu

**Keywords:** crowdsourcing, machine learning, diagnostics, telemedicine, autism, pediatrics

## Abstract

Mobilized telemedicine is becoming a key, and even necessary, facet of both precision health and precision medicine. In this study, we evaluate the capability and potential of a crowd of virtual workers—defined as vetted members of popular crowdsourcing platforms—to aid in the task of diagnosing autism. We evaluate workers when crowdsourcing the task of providing categorical ordinal behavioral ratings to unstructured public YouTube videos of children with autism and neurotypical controls. To evaluate emerging patterns that are consistent across independent crowds, we target workers from distinct geographic loci on two crowdsourcing platforms: an international group of workers on Amazon Mechanical Turk (MTurk) (N = 15) and Microworkers from Bangladesh (N = 56), Kenya (N = 23), and the Philippines (N = 25). We feed worker responses as input to a validated diagnostic machine learning classifier trained on clinician-filled electronic health records. We find that regardless of crowd platform or targeted country, workers vary in the average confidence of the correct diagnosis predicted by the classifier. The best worker responses produce a mean probability of the correct class above 80% and over one standard deviation above 50%, accuracy and variability on par with experts according to prior studies. There is a weak correlation between mean time spent on task and mean performance (*r* = 0.358, *p* = 0.005). These results demonstrate that while the crowd can produce accurate diagnoses, there are intrinsic differences in crowdworker ability to rate behavioral features. We propose a novel strategy for recruitment of crowdsourced workers to ensure high quality diagnostic evaluations of autism, and potentially many other pediatric behavioral health conditions. Our approach represents a viable step in the direction of crowd-based approaches for more scalable and affordable precision medicine.

## 1. Introduction

Autism spectrum disorder (ASD or autism) is a developmental delay with a continuously rising prevalence in the United States [1,2]. Access to care can be limited, particularly in rural and in lower socioeconomic areas, as families must wait for over a year to receive a formal diagnosis [3] and therefore treatment. Epidemiological estimates indicate that over 80% of U.S. counties do not have autism diagnostic resources [4]. Scalable and accessible tools would begin to address these inefficiencies in the healthcare system. Since autism consists of a largely behavioral phenotype, video data are a particularly powerful and rich means of capturing the range of social symptoms a child may exhibit in a fast and virtually cost-free manner. Accurate diagnoses and behavioral classifications have been inferred from categorical ordinal labels extracted by untrained humans from the short video clips [5,6,7,8,9,10], which are recorded by digital mobile and wearable interventions during use by the child or administering parent [11,12,13,14,15,16,17,18,19,20,21,22]. Such a process can be scaled through crowdsourcing platforms, which allow distributed workers from around the globe to perform short on-demand tasks.

Crowdsourcing offers a powerful mobilized telemedicine solution to providing a rapid and personalized diagnosis for and behavioral characterization of children at risk for developmental delays. Crowdsourcing is increasingly being used for sensitive work such as mental health tracking [23,24], body weight inference [25], prescription drug use and reactions [26,27], and investigating crime [28,29,30]. While at least partially automated diagnostics is an important goal for precision healthcare [31,32,33,34], the quantification and categorization of several social activities are beyond the scope of current machine learning methods [19], resulting in a major barrier in the field of precision medicine for behavioral conditions. While answers from a crowd worker on a variety of behavioral dimensions can provide a precision diagnosis for one individual, each label can be used as training data for a general-purpose machine learning model that can make precision medicine more automated and scalable. Achieving this goal, however, relies on high quality data from the crowd [35,36], necessitating careful characterization of worker performance and subsequent filtering of crowd workers.

Here, we evaluate the performance of individual workers within four independent pools of crowdsourced workers from Amazon Mechanical Turk (MTurk) [37,38], a popular paid crowdsourcing platform, and Microworkers [39,40], another paid crowdsourcing platform with a significantly larger international pool of workers compared to MTurk [41]. The workers watch unstructured videos of children with autism and neurotypical controls and fill out a series of multiple-choice questions about the child’s behavior. The series of multiple-choice answers serve as a vector of categorical ordinal features used as input to a previously validated [42,43] logistic regression classifier distinguishing neurotypical children from autistic children. We assess the differences in classifier probabilities and final predictions across workers, finding that there are significant differences in worker performance despite identical video difficulty levels. These results suggest that crowd workers must be filtered before incorporation into clinical diagnostic practices. 

## 2. Materials and Methods

Our methods consist of (1) first identifying a clinically representative video set, (2) choosing an appropriate classifier for evaluating worker responses, and (3) crowdsourcing a video rating task to a wide pool of global workers.

### 2.1. Clinically Representative Video Set

Clinically representative videos were downloaded from YouTube. We selected publicly available videos of both children with and without autism. Diagnosis was based on video title and description reported by the uploader. We only selected videos that matched all of the following criteria: (1) the child’s hands and face are clearly visible, (2) there are opportunities for social engagement, and (3) there is at least one opportunity for using an object such as a toy or utensil. No further selection criteria were used.

All three rating groups on Microworkers received the same set of 24 videos with a mean duration of 47.75 s (SD = 30.71 s). Six videos contain a female child with autism, six videos contain a neurotypical female child, six videos contain a male child with autism, and six videos contain a neurotypical male child. The mean age of children in the video was 3.65 years (SD = 1.82 years). 

We asked 4 licensed clinical experts (2 Child and Adolescent Psychiatrists, 1 Clinical Psychologist, and 1 Speech Language Pathologist) to watch each video of the 12 children with an autism diagnosis and to rate the severity of the child’s autism symptoms according to the first question of the Clinical Global Impression (CGI) [44] scale measuring the “severity of illness” between 1 (“normal, not at all ill”) to 7 (“among the most extremely ill patients”). We then recorded the mean rating rounded to the nearest whole number. There was one video with a mean rating of 2 (“borderline mentally ill”), four with a mean of 4 (“moderately ill”), five with a mean of 5 (“markedly ill”), and two with a mean of 6 (“severely ill”), validating that we posted a clinically representative set of videos on Microworkers.

We additionally conducted a post-hoc analysis of previously crowdsourced yet unpublished pilot test results from MTurk with the exact same rating tasks except using a separate set of 43 videos to rate with a mean duration of 43.85 s (SD = 26.06 s). Ten videos from this set contain a female child with autism, eleven videos contain a neurotypical female child, twelve videos contain a male child with autism, and ten videos contain a neurotypical male child. The mean age of children in the video set was 3.61 years (SD = 1.61 years). The 4 clinical experts rated the 22 children with autism in this set using the CGI. There were three videos with a mean rating of 2 (“borderline mentally ill”), five with a mean of 3 (“mildly ill”), three with a mean of 4 (“moderately ill”), six with a mean of 5 (“markedly ill”), and five with a mean of 6 (“severely ill”), validating that we posted a clinically representative set of videos on MTurk.

### 2.2. Video Observation Classifier

To evaluate the performance of crowd workers against a clinician gold standard, a previously validated binary logistic regression classifier [42,43] was trained on electronic health record data consisting of clinician filled Autism Diagnostic Observation Schedule (ADOS) [45] scoresheets for 1319 children with autism and 70 non-autism controls. We chose logistic regression over alternative classical machine learning techniques like support vector machines and alternating decision trees because of the previously published head-to-head comparison of these techniques by Tariq et al. [43], which found that logistic regression resulted in both the highest accuracy and highest unweighted average recall. We used the default *scikit-learn* parameters for logistic regression, except we evaluated both L1 and L2 regularization with an inverse regularization strength of 0.05, forcing strong regularization. We reported the metrics with the greatest accuracy of L1 or L2 regularization. Because our goal was to evaluate worker performance and not to maximize the performance of a classifier, we did not perform any further hyperparameter tuning.

Because logistic regression classifiers emit a probability for a binary outcome, we treat the probability as a confidence score of the crowdsourced workers’ responses. Here, we exclusively analyze the probability of the correct class (referred to as PCC from here on out), which is *p* when the true class is autism and *1-p* when the true class is neurotypical. When assessing classifier predictions, we use a threshold of 0.5. Throughout this paper, we refer to a worker’s average PCC for videos the worker rated as a measure of the worker’s video tagging capability. Similarly, we refer to a video’s PCC as the difficulty level of the video.

### 2.3. Video Rating Tasks

We aimed to crowdsource workers from three culturally distinct countries where autism is prevalent yet access to resources is lacking. These are samples of areas where accessible, affordable, and scalable precision medicine solutions, such as instantiations of the technique described here, can enable access to care to underserved populations globally. In particular, we selected Bangladesh, Kenya, and the Philippines, countries that collectively represent diverse areas containing problematic issues with autism prevalence and limited access to services [46,47,48,49]. 

In order to generalize our findings across these distinct groups of workers, we posted four sets of video rating tasks under the following hypotheses:

**H1a.** 
*There are workers on MTurk whose mean classifier PCC will exceed 75%.*


**H1b.** 
*There are “super recognizer” workers on MTurk whose mean classifier PCC will exceed 75% and whose mean will be over one standard deviation above 50%.*


**H2a.** 
*There are workers from Bangladesh on Microworkers whose mean classifier PCC will exceed 75%.*


**H2b.** 
*There are “super recognizer” workers from Bangladesh on Microworkers whose mean classifier PCC will exceed 75% and whose mean will be over one standard deviation above 50%.*


**H3a.** 
*There are workers from Kenya on Microworkers whose mean classifier PCC will exceed 75%.*


**H3b.** 
*There are “super recognizer” workers from Kenya on Microworkers whose mean classifier PCC will exceed 75% and whose mean will be over one standard deviation above 50%.*


**H4a.** 
*There are workers from the Philippines on Microworkers whose mean classifier PCC will exceed 75%.*


**H4b.** 
*There are “super recognizer” workers from the Philippines on Microworkers whose mean classifier PCC will exceed 75% and whose mean will be over one standard deviation above 50%.*


We evaluate the above hypotheses only for workers who rated at least ten videos and for videos that received at least ten sets of ratings from workers. Hypotheses H1a, H2a, H3a, and H4a verify that there exist workers whose mean classifier PCC is consistently higher than the classification decision boundary by a sizable margin (25%) that is consistent with the documented 75% agreement rate between qualified multidisciplinary team diagnosis [50] using the Autism Diagnostic Observation Schedule (ADOS) [45] and Gilliam Autism Rating Scale (GARS) [51] scales. Hypotheses H1b, H2b, H3b, and H4b are more stringent, requiring the worker to exhibit low enough variance in their answers such that one standard deviation below their mean PCC is still above the classifier decision boundary and therefore still yields the correct diagnostic prediction. This level of robustness to variability is reasonable given the measures of inter-rater reliability of ADOS scoresheets, with Cohen’s kappa coefficients for individual ADOS items ranging between 0.24 and 0.94 [52].

The first set of tasks was posted on MTurk. The second, third, and fourth sets were posted to distinct groups of workers on Microworkers. The Microworkers crowdsourcing tasks were targeted to workers in Bangladesh, Kenya, and the Philippines in order to sample a sufficiently diverse global population of crowdworkers while comparing independent subsets of the crowd. 

All four independent studies consisted of a series of 13 multiple choice questions which were fed as inputs into the video observation classifier (Figure 1). Although the videos did not necessarily contain evidence of all 13 behavioral features used as inputs, workers were asked to infer how the child would behave if placed in the situation in question. 

On MTurk, workers were only allowed to proceed with rating further videos if they passed a series of quality control metrics recording performance against the ADOS gold standard classifier (see section Materials and Methods: Video Observation Classifier) and the time spent working on the task. On Microworkers, worker filtering did not occur besides requiring a bare minimum of time rating each video (a minimum of 2 min per video was required to accept a worker’s response). 

## 3. Results

We analyze (1) the distribution of worker performance in different countries and crowd platforms, (2) the number of higher performing workers and “super recognizers” in each study group, and (3) the correlation between mean time spent on the task and mean worker performance for each study group.

### 3.1. Distribution of Worker Performance

For all four worker groups, there was major variation in the average probability score of the classifier per video (Figure 2) and per worker (Figure 3). The mean probability of the true class for the 43 videos with at least ten worker ratings on MTurk was 63.80% (SD = 13.78%), with a minimum of 16.90% and a maximum of 84.05%. On Microworkers, the mean PCC for videos with at least ten ratings were 63.15% (N = 24; SD = 10.42%; range = 33.73–79.03%) for Bangladesh, 67.75% (N = 24; SD = 14.68%; range = 32.71–88.91%) for Kenya, and 72.05% (N =2 4; SD = 13.05%; range = 47.84–90.80%) for the Philippines. 

The mean classifier PCC for the 15 workers with 10 or more videos rated on MTurk was 66.67% (SD = 5.98%), with a minimum of 48.16% and a maximum of 70.82%. On Microworkers, the mean worker classifier PCC for those who provided ten or more ratings were 62.53% (N = 56; SD = 10.43%; range = 42.21–79.53%) for Bangladesh, 66.67% (N = 23; SD = 9.75%; range = 46.62–81.03%) for Kenya, and 72.71% (N = 25; SD = 10.25%; range = 51.38–89.08%) for the Philippines. 

Crucially, while there were individual differences in the subset of videos rated across workers, there was no significant difference in the difficulties of these videos across workers (Figure 4) for workers who rated at least ten videos and videos with at least ten ratings from workers. This confirms that the variability in worker performance in not attributable to the video difficulties. In all four groups, a Pearson correlation test between the mean video PCC and mean worker PCC yielded insignificant results (*r* = 0.088, *p* = 0.07 for MTurk; *r* = 0.017, *p* = 0.62 for Bangladesh Microworkers; *r* = −0.018, *p* = 0.70 for Kenya Microworkers; and *r* = −0.030, *p* = 0.52 for Philippines Microworkers).

### 3.2. Super Recognizers

Figure 3 and Figure 4 reveal that hypotheses H1a, H2a, H3a, and H4a are confirmed: there were workers in all four study groups whose mean classifier PCC exceeded 75%. On MTurk, there was one worker whose mean was greater than one standard deviation above 50%, confirming hypothesis H1b. There were three Microworkers in the Bangladesh cohort, two Microworkers in the Kenya cohort and ten Microworkers in the Philippines cohort whose mean was greater than one standard deviation above 50%, confirming hypotheses H2b, H3b and H4b.

### 3.3. Effect of Time Spent Rating

Because of pervasive practices among MTurk workers of artificially inflating the time spent on the task out of fear of spending insufficient time on the task [53,54], we only analyzed timing information of Microworkers data. Several recorded times for MTurk tasks exceeded several hours, suggesting MTurk worker behavior of bloating task times.

There was no statistically significant Pearson correlation between mean time spent on the task and mean worker performance for the Kenya and Philippines Microworkers groups individually (*r* = 0.191, *p* = 0.38 for Kenya Microworkers; and *r* = 0.193, *p* = 0.35 for Philippines Microworkers). For Bangladesh Microworkers, there was a statistically significant correlation (*r* = 0.326, *p* = 0.01). When aggregating all Microworkers results, the correlation is slightly strengthened (*r* = 0.358, *p* = 0.005).

## 4. Discussion

We discuss (1) the overall implications of the worker variability in all study groups and the presence of “super recognizers,” (2) the formalization of a crowd filtration process which can be leveraged for the identification of high performing crowd workers for a variety of precision medicine tasks, and (3) limitations of the present study and areas of potential future work.

### 4.1. General Implications

All four independent worker groups produced at least one worker who rated at least ten videos and whose mean classifier PCC exceeded 75%. There was one MTurk worker, three Microworkers in Bangladesh, two Microworkers in the Kenya cohort, and ten Microworkers in the Philippines cohort whose mean was greater than one standard deviation above 50%. It is unclear whether language barriers, differences in Microworker demographics across countries, or other factors are responsible for this inconsistency across countries.

We observe a high variation in worker performance in all four study groups. This variation in performance is distinct from other common crowdsourcing tasks such as image labeling, where worker responses are generally accepted to be high quality and therefore only simple quality control metrics (rather than filtering processes) are usually in place. These results suggest that there are innate differences between crowd workers’ abilities to successfully and accurately label behavioral features from short unstructured videos of children. This variation in intrinsic ability to rate diagnostically rich features suggests that a filtering process must occur to curate a subset of the crowd who are skilled at inferring behavior patterns from videos. We term this skilled distributed workforce “super recognizers” as they appear consistently adept at recognizing and tagging core autism symptoms from unstructured video without prior training. 

Further, we find that the time spent rating is weakly correlated with average performance, indicating that workers can be filtered for spending too little time on the tasks in aggregate. Although this trend was not observed in the Kenya and Philippines cohorts individually, this may likely be attributed to the smaller sample sizes of these groups. Including these data in the aggregate time correlation analysis bolstered the statistical significance of the correlation.

Gold standard classifiers trained on clinician-filled electronic health records are pertinent to scaling digital behavioral diagnostics. The source of training data is crucial, as behavioral instruments are not always consistent with categorizing diagnostic outcomes for the same individual [55]. The classifier used here was trained on ADOS records, but the children in the videos were not necessarily diagnosed via the ADOS, as there are several diagnostic instruments for autism [56,57,58,59,60,61] capturing overlapping yet distinct behaviors.

It is clear that different workers possess varying capabilities in behavioral video tagging, a nontrivial task. To realize economic crowdsourcing, several subsets of the crowd should be evaluated, with adept subgroups further pursued. Curating such a skilled crowd workforce in a developing country may lead to part time employment of “super recognizers” in telemedical practices in that country. This would eventually enable automated precision medicine through training machine learning classifiers using the labeled video data libraries accumulated through distributed behavioral video tagging.

### 4.2. Formalization of a Crowd Filtration Process

These results suggest that clinical workflows incorporating crowdsourced workers for pediatric diagnostics of complex behavioral conditions like autism should first filter down the crowd to a subset of workers who repeatedly and consistently perform well. Here, we propose a novel workflow for recruitment of crowdsourced workers to ensure high quality diagnostic evaluations of pediatric behavioral health conditions:Train one or more machine learning classifiers using data accumulated by domain expert clinicians. These data may be actively acquired or mined from existing data sources. It is crucial that the gold standard data are representative of the target pediatric population.Define a target performance metric for worker evaluation and a target number of workers to recruit.Collect labels from a massive and distributed set of crowd workers (Figure 5).Filter the crowd workers progressively and repeatedly until the target number of workers have reached or surpassed the target performance metric.The final set of globally recruited “super recognizers” can be leveraged in precision health and precision medicine clinical workflows toward rating a worldwide pediatric population (Figure 5).

### 4.3. Limitations and Future Work

There are several limitations of the present study and fruitful avenues for future work. More structured videos, such as those collected in home smartphone autism interventions [15,16,17,18,19], may yield more consistent video difficulty levels due to the standardization of collected videos. Mobile therapeutics in conjunction with crowdsourcing may be leveraged toward longitudinal outcome tracking of symptoms [7]. Testing more subsets of the crowd, partitioned not only by location but by a wide array of demographic factors, will reveal economical subsets of the crowd for remote behavioral video tagging. To understand the reasons for differences in performance across subsets, videos of children that reflect the demographics of the population being targeted should be deployed and compared against a control set of videos. We welcome and call for replication crowdsourcing studies with separate video sets and crowd recruitment strategies. We also hope similar approaches to those tried here will be replicated for other behavioral conditions such as ADHD, speech delay, and OCD.

## 5. Conclusions

Crowdsourcing is a powerful yet understudied emerging tool for telemedical precision medicine and health. We have demonstrated that crowdsourced workers vary in their performance on behavioral tagging of clinically representative videos of autism and matched neurotypical controls, and we provide formalization of a crowd filtration process for curation of the most capable members of the crowd for repeated use in crowdsourcing-based clinical workflows. This process consists of training a classifier from records filled by domain experts, identifying quantitative metrics for evaluating workers, crowdsourcing a clinical task, filtering workers using the clinician-trained classifier, and repeating until the ideal workforce size has been reached.

As data from human crowd-powered telemedical precision medicine pipelines are recorded and stored in growing databases, computer vision classifiers of core autism symptoms such as hand stimming, eye contact, and emotion evocation can be trained using these labeled datasets. Curation of a workforce of “super recognizers” will allow clinicians to trust the diagnostic labels and allow engineers to use high quality features when training novel classifiers for precision medicine. This will enable an eventual increase in the automation, and therefore throughput, of precision medicine techniques for pediatric developmental delays such as autism. 

## Figures and Tables

**Figure 1 jpm-10-00086-f001:**
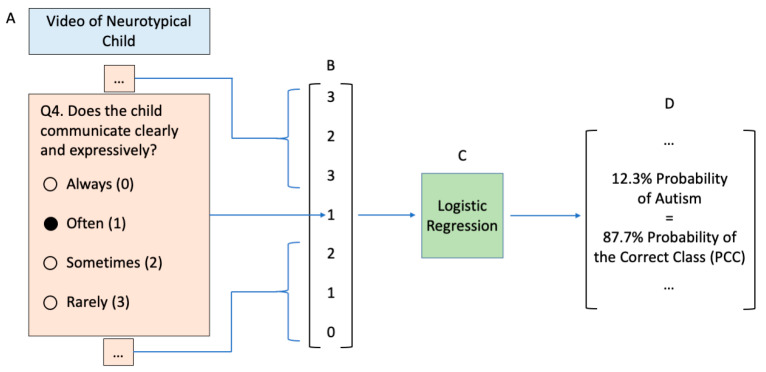
The process for calculating a probability score of autism from the categorical answers provided by crowdsourced workers. (**A**) Workers answer a series of multiple-choice questions per video that correspond to (**B**) categorical ordinal variables used in the input feature matrix to the (**C**) logistic regression classifier trained on electronic medical record data. This classifier emits a probability score for autism, which is the probability of the correct class when the true class is autism and 1 minus this probability when the true class is neurotypical (the latter case is depicted). (**D**) A vector of these probabilities is used to calculate mean worker and mean video probabilities of the correct class.

**Figure 2 jpm-10-00086-f002:**
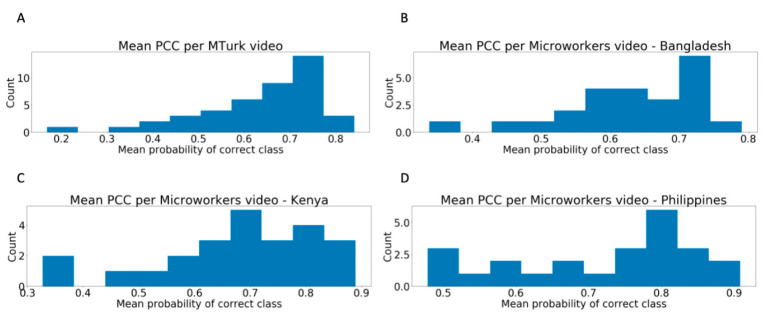
Distribution of average classifier probability of the correct class per video with at least ten ratings from (**A**) MTurk workers, (**B**) Bangladesh Microworkers, (**C**) Kenya Microworkers, and (**D**) Philippines Microworkers. There is wide variability in the difficulty level of rated videos.

**Figure 3 jpm-10-00086-f003:**
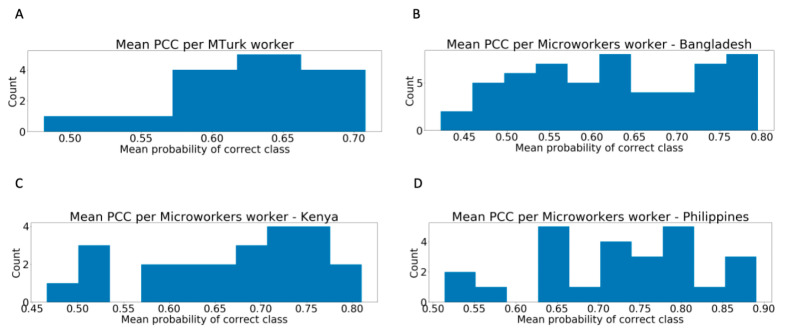
Distribution of average probability of the correct class per (**A**) MTurk worker, (**B**) Bangladesh Microworker, (**C**) Kenya Microworker, and (**D**) Philippines Microworker who provided at least ten ratings. There is wide variability in the ability of workers to provide accurate categorical labels.

**Figure 4 jpm-10-00086-f004:**
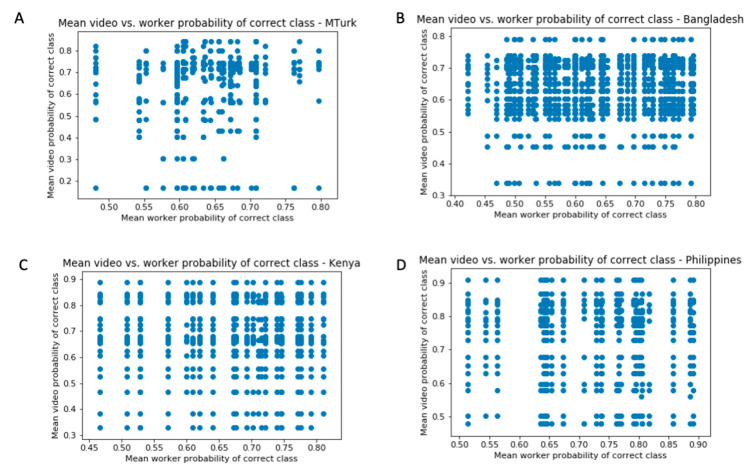
Mean classifier confidence per video vs. per worker for (**A**) MTurk workers, (**B**) Bangladesh Microworkers, (**C**) Kenya Microworkers, and (**D**) Philippines Microworkers for videos with at least 10 ratings and workers who provided at least ten ratings. Each vertical line of points contains the difficulty levels of videos rated for one worker, visually demonstrating that workers received similar distributions of video difficulties to rate despite displaying large variation in average diagnostic confidence.

**Figure 5 jpm-10-00086-f005:**
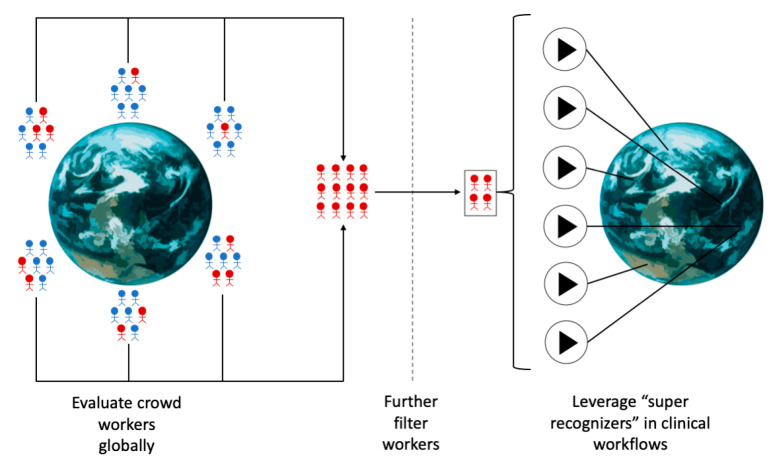
Crowd filtration pipeline. Crowdsourced workers are first evaluated globally. The highest performers from each location are further evaluated for one or more rounds until a final skilled workforce is curated. These “super recognizers” may then be repeatedly employed in global clinical workflows.

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
