# Peer review of "Precision Telemedicine through Crowdsourced Machine Learning: Testing Variability of Crowd Workers for Video-Based Autism Feature Recognition"

_jpm, 2020, doi:10.3390/jpm10030086_

Round 1

Reviewer 1 Report

The article has a correct IMRaD structure. The section references has a proper number of positions and references are modern. The introduction state the purpose of the paper. The figures have a proper aesthetics but Authors should enlarge descriptions of figures 2 and 3.The article is written rationally. The article is interesting and valuable. The main novelty in the work is using crowdsourcing in order to give an answer about ASD which is important problem in the U.S.A.. The article is important to colleagues working in the field. Binary logistic regression classifier is used correctly in the article. Conclusions are given correctly. Video data can be used in order to find behavioral autism features in children. You should elongate the conclusions section.

Author Response

We would like to thank Reviewer 1 for your helpful comments. We have revised our manuscript by addressing all comments.

The figures have a proper aesthetics but Authors should enlarge descriptions of figures 2 and 3.

We have increased the descriptions in Figures 2 and 3 to be much more readable by doubling the font size and shortening the text. For Figure 2, the titles were shortened from “Distribution of mean probability of correct class for each video on [platform] – [country]” to “Mean PCC per [platform] video – [country]”. For Figure 3, the titles were shortened from “Distribution of mean probability of correct class for each [platform] worker – [country]” to “Mean PCC per [platform] worker – [country]”.

You should elongate the conclusions section.

We have added to the Conclusions section. Starting on line 328, we summarize the steps of the process for quantitatively and systematically recruiting a diagnostic workforce from the crowd. Starting on line 333, we elaborate on the potential use cases of our technique in the real-world practice of precision medicine.

Reviewer 2 Report

The authors address a very relevant issue and propose to apply machine learning techniques to recognize autism on the basis of data coming from crowdsourcing environments.

The authors illustrate how they analyzed the problem and propose a first solution that uses videos coming from YouTube and take advantage of rated workers.

The approach proposed in the paper gives the bases for further work and investigations in the area and can be considered as a preliminary study: lot of work can be still done, as underlined also by the authors in the conclusions of the paper.

The paper is well structured but a more detailed description of the tools and of the implemented software could be useful to the readers.
Not enough detail is given about the chosen classifier. Did the authors try any alternative? How was it chosen? How is it working? Can be parameterized?
Moreover I suggest to add a small introduction to each section. In the current version of the paper there is no text between any section and any following subsection (i.e., between Section 2 and subsection 2.1, between Section 3 and subsection 3.1). I would suggest to add a little summary of the content of each Section.

Author Response

We would like to thank Reviewer 2 for your helpful comments. We have revised our manuscript by addressing all comments.

The paper is well structured but a more detailed description of the tools and of the implemented software could be useful to the readers. Not enough detail is given about the chosen classifier. Did the authors try any alternative? How was it chosen? How is it working? Can be parameterized?

We have detailed information about the logistic regression classifier starting in line XXX. In particular, we add the following text: “We chose logistic regression over alternative classical machine learning techniques like support vector machines and alternating decision trees due to the previously published head-to-head comparison of these techniques by Tariq et al. [42], which found that logistic regression resulted in both the highest accuracy and highest unweighted average recall. We used the default scikit-learn parameters for logistic regression, except we evaluated both L1 and L2 regularization with an inverse regularization strength of 0.05, forcing strong regularization. We reported the metrics with the greatest accuracy of L1 or L2 regularization. Because our goal is to evaluate worker performance and not to maximize the performance of a classifier, we did not perform any further hyperparameter tuning.”

Moreover I suggest to add a small introduction to each section. In the current version of the paper there is no text between any section and any following subsection (i.e., between Section 2 and subsection 2.1, between Section 3 and subsection 3.1). I would suggest to add a little summary of the content of each Section.

We have added a small introduction and summary to each section and subsection. These can be found starting in lines 80, 181, and 251.